# Prediction of Silicon Content in a Blast Furnace via Machine Learning: A Comprehensive Processing and Modeling Pipeline

**DOI:** 10.3390/ma18030632

**Published:** 2025-01-30

**Authors:** Omer Raza, Nicholas Walla, Tyamo Okosun, Kosta Leontaras, Jason Entwistle, Chenn Zhou

**Affiliations:** 1Center for Innovation through Visualization and Simulation (CIVS) and Steel Manufacturing Simulation and Visualization Consortium (SMSVC), Purdue University Northwest, Hammond, IN 46323, USA; raza12@pnw.edu (O.R.); njwalla@pnw.edu (N.W.); czhou@pnw.edu (C.Z.); 2U. S. Steel Gary Works, 1 Broadway, Gary, IN 46402, USA; kleontaras@uss.com (K.L.); jwentwistle@uss.com (J.E.)

**Keywords:** blast furnace, hot metal chemistry, silicon prediction, machine learning, XGBoost

## Abstract

Silicon content plays an important role in determining the operational efficiency of blast furnaces (BFs) and their downstream processes in integrated steelmaking; however, existing sampling methods and first-principles models are somewhat limited in their capability and flexibility. Current data-based prediction models primarily rely on a limited set of manually selected furnace parameters. Additionally, different BFs present a diverse set of operating parameters and state variables that are known to directly influence the hot metal’s silicon content, such as fuel injection, blast temperature, and raw material charge composition, among other process variables that have their own impacts. The expansiveness of the parameter set adds complexity to parameter selection and processing. This highlights the need for a comprehensive methodology to integrate and select from all relevant parameters for accurate silicon content prediction. Providing accurate silicon content predictions would enable operators to adjust furnace conditions dynamically, improving safety and reducing economic risk. To address these issues, a two-stage approach is proposed. First, a generalized data processing scheme is proposed to accommodate diverse furnace parameters. Second, a robust modeling pipeline is used to establish a machine learning (ML) model capable of predicting hot metal silicon content with reasonable accuracy. The method employed herein predicted the average Si content of the upcoming furnace cast with an accuracy of 91% among 200 target predictions for a specific furnace provisioned by the XGBoost model. This prediction is achieved using only the past shift’s operating conditions, which should be available in real time. This performance provides a strong baseline for the modeling approach with potential for further improvement through provision of real-time features.

## 1. Introduction

Silicon content in hot metal is deeply intertwined with the operational efficiency and quality assurance of blast furnaces (BFs). The BF process involves injecting hot gas that reduces the iron ore in the burden descending from the top of the furnace. The molten products (generally referred to as hot metal) are collected at the furnace hearth along with molten oxides, which form a second liquid phase of slag. Hot metal and slag are tapped from the furnace hearth and diverted into separate streams for steelmaking and other processing. This complex system necessitates a sensitive balance for optimal product yield and quality [1], and it demands careful control over internal conditions, further compounded by the needs for improved energy efficiency and emissions control [2].

The level of silicon dissolved in the hot metal, introduced as part of charged raw materials or fuels, is an important quality metric and often also positively correlates strongly with the energy present in the high-temperature regions of the furnace. Low or falling Si levels may warn operators of approaching stability concerns such as abnormal nodulation, excessive skull formation, and potentially even industrial accidents. Rising or high Si levels may indicate unnecessary heat generation (through wasted coke combustion) and poor energy efficiency, in addition to adverse downstream impacts [3]. Ascertaining Si content, therefore, has significant implications for energy efficiency, environmental impact, and process economics. Furthermore, through these correlations, Si content also has implicit links with other blast furnace parameters and conditions such as fuel ratios, coal injection amount, air volume, air temperature, coke load, and smelting slag. Since the hostile environment within the furnace precludes direct temperature measurements, silicon content in the hot metal in the hearth emerges as a key proxy measure, offering a tangible link to furnace conditions and health.

However, to determine silicon content, significant time delays are present in the response of the blast furnace to operator inputs when looking at hot metal composition and temperature measurements, as well as in the lab analysis of such measurements, presenting challenges in the use of direct measurements for real-time control. Consequently, first-principles models, coupled with domain knowledge of heat conduction, energy, and chemical equations, have been deployed to infer silicon content. However, this approach can require significant computation time and may not be accurate enough to deterministically model and account for the complex range and magnitude of processes taking place—such as transient conditions in the high temperature furnace zones. Additionally, drivers with inherent stochasticity and a lack of detailed information exist. These include, but are not limited to, unknowns and uncertainties in charged material composition, transient conditions in the high-temperature zones of the furnace, and other influencing elements that are not easily captured in static models.

While the fundamental mechanisms behind the process have been established, the limiting factors established here encourage BF operators to explore methods that can predict silicon content in advance. To address this challenge, various data-driven models have been developed and consequently share a rich history of implementation alongside these physics-based models. These approaches involve sophisticated data pre-processing, feature selection, and optimization techniques [3,4,5], as demonstrated by numerous researchers [6,7,8,9]. They range from traditional statistical methods to contemporary machine learning approaches, which seek to improve accuracy and speed as their prime objectives.

Among earlier works, Liu employed Bayesian networks, while Gao explored chaos theory to develop a chaotic prediction method around 2005 [10,11]. Jian utilized radial basis functions and support vector machines (SVMs), and Gao applied fuzzy SVMs for control limit determination around 2008. Bhattacharya employed partial least squares, while Liu and Zeng used dimensionality reduction techniques such as principal component analysis around 2009. In 2011, Jian adopted the smooth SVR method for predicting silicon content in molten iron. Time series analysis methods, proposed by Waller and Saxén [12], were also explored. Although these traditional methods showed promise and decent accuracy under specific conditions, they struggled to incorporate large datasets or capture the complexity of reactions influencing silicon yield.

As machine learning became more feasible, more complex models gained traction. Wang improved predictions in 2014 using random forests with numerous features. In 2015, Particle Swarm Optimization with SVRs enhanced convergence speed and parameter optimization [13]. By 2017, shallow neural networks such as Extreme Learning Machines (ELMs), combined with outlier detection and feature selection, achieved notable accuracy [14,15]. Later, XGBoost and LightGBM outperformed traditional models, while simple Multi-Layer Perceptrons (MLPs) proved effective, lightweight, and fast. Techniques such as Gray Relational Analysis and fuzzy clustering, integrated with neural models, enabled effective feature selection and identification of time lags for dynamic predictions under fluctuating furnace conditions [16,17]. Overall, MLPs and deep neural networks further improved accuracy with significant speed gains.

However, most of these works still do not incorporate an expansive set of furnace parameters and state variables, only a select few. One important reason is the overhead involved in establishing automated data processing pipelines. While domain knowledge necessitates manual intervention and judgment, many features could be automatically processed and filtered, potentially leading to better accuracy. As a result, there exists a rich history of predictive modeling work that spans traditional statistical models and contemporary machine learning models. These models have achieved good accuracy and speed but often focus on a specific subset of features, typically requiring significant manual deliberation, which may limit further improvement in accuracy. A similar practice is observed in predicting other industrial process variables, such as carbon brick temperature [18], in addition to silicon content.

The limitations of previous works are addressed through the objectives and contributions of this current effort, which are twofold: (1) a generalized yet adaptable automated data processing pipeline and (2) an effective ML modeling, tuning, and selection framework. Existing modeling approaches in the literature often lack a comprehensive and generalized data processing pipeline. Such a pipeline is essential to lower data requirements and costs, reduce corrupted inputs, accelerate convergence, and improve model accuracy. Effective data processing, particularly in identifying anomalies and removing collinearities, is arguably more critical than modeling in physical processes. This is because real-time process data often lack the accuracy and standardization typically present in preprocessed data used in ML-driven research. Consequently, the greatest performance gains may stem from robust data processing. Furthermore, the pipeline needs to be as model-agnostic as possible to support diverse model training and selection, enabling the identification and fine-tuning of the best predictive models for various purposes, such as silicon content prediction [17] Currently, the absence of a comprehensive yet versatile data processing pipeline, combined with the lack of a broad modeling framework, limits the accuracy and broader adoption of data-driven modeling for silicon content prediction. Our pipeline addresses these gaps by integrating both data processing and modeling modules to streamline the predictive process, making it applicable across different sites and process variables and improving predictive capabilities. Improved accuracy in predicting silicon content can help furnace operators adjust parameters effectively, thereby maintaining production yield while reducing operating time, cost, and energy consumption.

## 2. Methodology

### 2.1. Data Overview

#### 2.1.1. Data Acquisition and Characteristics

Multiple tabular data sources for a blast furnace of typical size and productivity were provided by a major North American steel manufacturer for training and testing purposes with varying time-based granularity and intervals. There were approximately 800 data features—variables representing furnace parameters or state—across all the data sources, necessitating proper processing before data could be fed into the model. Features included conditions such as blast temperature; uptake gas temperature; gas pressure; thermocouple temperatures; chemical composition of charged burden materials, liquid metal, and collected gases; and burden charge weights, among other data. These data features could generally be categorized into four groups: input control parameters (such as blast pressure), furnace state conditions (such as uptake gas temperature), metadata (such as casting taphole identifier), and output variables (silicon content in our case, but other cases can include other output variables). The data types were mostly numerical for different tables, around 720, followed by 60 categorical and 6 timestamps, with the remaining being metadata strings. The average silicon content of hot metal for a given cast is the prime output variable with which this effort is concerned, but correlations may exist with other related output variables, such as the fraction of silicon in slag versus hot metal. Similarly, furnace state conditions such as gas uptake pressure can be influenced via control parameters such as blast pressure, albeit not orthogonally and with potential time lags. A selective but diverse range of features and the output silicon content %, with relevant statistics, such as mean, standard deviation, and interquartile range, are shown in Table 1. These features encompass and provide a basic overview of the parameters, conditions, and state of the furnace.

#### 2.1.2. Pipeline

To infer data characteristics; address data integrity, linkage, interdependencies, and time lag issues; and finally obtain the best predictive model, a generalized yet flexible pipeline has been established. Figure 1 provides a high-level overview of our pipeline, which includes comprehensive data processing and modeling techniques, adaptable for various industrial modeling or data processing purposes. The data processing stage of the pipeline allows an automatic way, with user input parameters, to link, format, and clean the data in the first step. Afterwards, it provides basic data parameters such as the mean and standard deviation, which can be used to provide visuals for manual inspection or allow automatic filtering. Following this, complex anomaly checks and collinearity checks are conducted to infer the most relevant features and their lags to use for prediction. Finally, these features are standardized. The modeling stage uses the processed data to infer the best hyperparameters. These candidate models are further cross validated to provide a final model that yields the best silicon content predictions.

Mathematically, with a degree of non-strictness, the processing and modeling transformations can be formulated as follows: let R be the tabular data matrix, with row index i, R_i_, and column index j, R_j_. The rows indicate a certain time, event, or entity, such as date or cast, containing information pertinent to furnace conditions. Columns indicate the furnace parameters such as blast pressure. Thus, some of the important data transformations are as follows:(1)f_formatting_(R) → R’ where R_a,__ ≠ R_b,_ for any a ≠ b and likewise for the column.(2)f_anomaly_(R) → M where M_i,j_ = 1 if R_i,j_ > anomaly threshold and 0 otherwise. M is a mask.(3)f_colinearity_(R__,A_) → R__,B_’ where B ⊆A.

The processed data is represented as X^t^ where index t indicates the furnace conditions at time t. For modeling, the overarching prediction process is as follows:

f_model_((X^t^,X^t−1^_subset_,X^t−2^_subset_,…X^t−r^_subset_)) → *y*, where r is the maximum lag determined by the correlation method and y is the predicted silicon content % for the next cast given these input furnace conditions. 

Overall, the data processing stage makes it possible to obtain a clean, standardized subset of the most relevant features, refined to about 50 features from about 800 features initially. The modeling stage tunes and chooses the best model based on the input parameters. These parameters, after due processing and selection, are used at the modeling stage with unavailable parameter values dealt accordingly.

### 2.2. Data Processing

#### 2.2.1. Data Formatting and Clean-Up

The most basic tasks involve formatting the tabular data so that string data, including column names, are standardized. Duplicates and invalid entries, based on data type inference, e.g., whether a feature consists of string, time, date, numerical or categorical, are also removed or masked. Table 2 shows the initial data inference stage results for the different data features from multiple data sources. Samples of data are provided alongside the resulting inferred type from our algorithm. Figure 2 outlines how our algorithm infers the different data types, with user-configurable parameters such as set thresholds for unique and numerical frequency, which can help determine whether a feature is a number or categorical. Date regex, i.e., different date patterns and formats, can help parse date and time data into timestamps.

#### 2.2.2. Basic Exploratory Data Analysis (EDA)

Despite the significant number of features in the data, it is still worthwhile to perform basic EDA. This involves analyzing basic data statistics such as mean, standard deviation, and common percentiles. Additionally, it is often useful to plot the distributions of data both in terms of values, e.g., histograms, and trends, as shown in time-series plots, to visually spot patterns. For instance, in our approach, a pattern can be noted where certain values, such as cast average titanium, are strictly zero as illustrated in Figure 3 where they appear skewed on the histogram. This indicates not that that cast average titanium percentage in the cast is actually zero but rather that such information is unknown, due to either sensor issues or data misplacement. Since the data are on an industrial scale, it is not always feasible for the data providers to keep track of and delineate such cases. Thus, a basic EDA can help promptly identify such cases where domain knowledge supersedes statistical observation. By removing or marking such values as invalid, we obtain the actual distribution of those data, which is unimodal and roughly follows a beta distribution, a common pattern for fractional industrial data yields [19] such as titanium percentage in our case. Visual inspection and guided beta fitting help identify these invalid values. Furthermore, basic statistics regarding each feature are obtained to attain a general sense of the data and an overview of potential anomalies. Such statistics, presented in Table 2, are used further downstream for anomaly detection purposes.

#### 2.2.3. Data Linkage, Collation, and Domain Knowledge-Based Sifting

After the basic formatting, clean-up, and EDA, individual data sources must be combined. Examples of such separate sources of information provided by the sensor readings and laboratory analysis include thermocouple temperatures sampled in minutes or hours, hot metal chemistry sampled by cast, and gas pressure values sampled by minute. To provide a comprehensive picture of all the furnace parameters and state variables during a given cast, these data sources must be combined. This also helps with analysis and downstream processing. For instance, the multicollinearity and VIF step, discussed later, can achieve much better feature reduction performance if it has access to all features rather than a subset of features at a time.

The data sources are linked to each other primarily via different configurable variables, such as timestamp, cast number, or ladle number, as outlined in Figure 4. An algorithm was prepared to combine different data sources based on configurable linking features. For features and events such as casting, the cast numbers and ladle numbers are matched exactly, as they are uniquely identifiable. However, the challenge was in combining features with different time granularity. As examples of such features, hearth bottom thermocouple values were provided on a by-hour basis, while hearth wall thermocouple and gas pressure values provided on a by-minute basis, and chemical composition values were averaged over each cast, spanning a few hours. To address this issue, the timestamps are matched based on closest backward proximity. For example, if a timestamp for a parameter set, p1, is 01:45 (per minute) while that of another parameter set, p2, has timestamps of 01:00 and 02:00 (per hour), p1 and p2 are matched at timestamps 01:00 and 01:14, hence the name closest backward proximity. At other times, because some sensor readings are frequent but do not vary significantly, redundancy is removed by replacing those frequent values with an average or median for that interval, which are plausible statistics describing the numbers in the interval. For instance, if blast pressure readings, per minute, are within a small range (perhaps 18.20–18.28), the granularity can be reduced by averaging these values over an hour. This reduces the size of the processed data and speeds up subsequent processing and analysis. The time granularity was chosen to be sufficiently indicative of their intervals to strike a balance between brevity and information retention. Typically, granularity is reduced to an hourly or per cast basis.

Finally, the variables (columns in the table) that are highly irrelevant to silicon content prediction are manually removed as part of the linkage process. This mainly includes metadata and helps reduce the features from 800 to approximately 750. This is the only fully manual aspect in the proposed pipeline, though it is not strictly necessary because most features can be removed automatically in the subsequent multicollinearity step. By the end of this step, a linkage has been established among all the disparate data sources so that all of the unified features are associated with consistent operational events such as casting or a single timepoint (or small interval).

#### 2.2.4. Data Anomaly Detection

With the association of all furnace parameters and state variables established, various filters and models are applied to remove outliers and noise. Simple filters involve checking expected upper bounds and lower bounds with thresholds obtained from domain knowledge. For instance, domain knowledge dictates that thermocouple temperatures in the hearth walls and hearth bottom, should be above room temperature and below 2200 °F.

More statistically complex models involving percentile thresholding and deviation thresholding coupled with time-series residuals, are also used to detect spikes rather than solely using fixed thresholds. This approach is referred to herein as dynamic thresholding. Specifically, the difference is considered between the 50th percentile and the 10th, v_10_, and 90th, v_90_, percentile respectively. Thus, the dynamic upper threshold is v_90_ + α × v_90_ − v_50_), where α can be tuned (α = 2 in this case). The lower threshold is, similarly v_10_ − α × (v_50_ − v_10_).

To illustrate the effect of these filters, a sample of thermocouple readings and their detected anomalies are presented in Figure 5. Obvious negatives as well as very low (below 50) and very high values (above 2200) are readily detected. However, there are also a few detected anomalies within these ranges, such as a 1500 °F point, which are in fact spikes. This detection is based on the combination of both dynamic thresholding and time-series residuals. Time-series residuals provide local variations, while dynamic thresholding of those variations determines typical deviations and thus anomalous deviations. It should be noted that the threshold here is intentionally relaxed as it is difficult to ascertain if such spikes are due to actual operational data or high sensor noise. Thus, the approach here seeks to minimize false positives. These methods and their workings are summarized in Figure 6.

Additionally, more complex anomaly detection methods are employed on correlated feature subsets, e.g., hot metal chemistry elements. These methods can model joint distributions rather than filtering each feature individually Specifically, we use Gaussian Mixture Models (GMMs), which fit the mean and standard deviation multidimensional parameters to infer the joint sample probability: the less probable samples are considered anomalous. We also use Isolation Forest (ISF) Models [20], which uses the number or depth of conditions required to isolate and identify a sample from its cluster as the anomaly score: the lesser the conditions to isolate a sample, the more unique a sample is, making it more anomalous. The data features used for GMMs and ISFs are focused on various chemical compositions, such as the composition in terms of the amounts of copper, manganese, titanium, etc., in the liquid metals. Only a select few features, constituting the subset, are possible at a time. Otherwise, the dimensions would be large (dimensionality curse), and these anomaly detection algorithms would either be numerically unstable or require a much larger dataset to be effective.

#### 2.2.5. Multicollinearity Analysis and Target Leakage

Multicollinearity analysis, or feature correlation, refers to ascertaining which input features—furnace parameters and state variables in our case—are correlated with one another. Removing such correlations reduces the parameter space size, thus speeding up subsequent processing, and increases model stability, thus lowering the chances of an overfitted, unreliable model. Additionally, it retains interpretability of features as the final set of features are a subset of the original, as opposed to other dimensionality reduction techniques such as PCA. It is important to remove anomalies prior so that those outliers do not interfere with establishing correlation, or lack thereof.

To perform this reduction, a two-way process is employed. First, a simple feature correlation map is generated on related subset of features, (e.g., hearth wall thermocouples, hot metal chemistry, gas pressure). This provides a visual cue of the potentially undesirable dependencies between some input features. For instance, Figure 7 shows a sample of highly correlated input feature subset, namely, hearth wall thermocouples. These thermocouples are placed in concentric circular layers at a particular height—“hw” indicates hearth wall while “hs” indicates hearth shell, s1–s8 indicates clockwise angle orientation, and “1” in “hw1” indicates the height level. Apart from obvious self-correlations, thermocouple triplets (thermocouples along the same radial direction) are highly correlated, as indicated by the 3 × 3 miniature squares in the figure. This makes sense as theoretically, the temperature sensor readings at close proximity in a straight radial direction have a linear relationship—approximated from log linear relationship of heat transfer in a cylinder. Finally, an appropriate threshold for the correlation values can be set to remove those features with high correlation.

After reducing each feature subset individually, a Variance Inflation Factor (VIF) can then be applied to the combined remaining parameter space. The VIF is a common procedure that employs regression coefficient scores of variables to determine correlation. It is, however, computationally intensive, hence the need to perform it after the simpler feature subset reduction in the first part. Overall, these steps reduce the number of features from 750 to approximately 140.

As a corollary, a target leakage test is performed, which is a collinearity test between input variables and the output variable, instead of with each other. A very high correlation here, for industrial data, can indicate presence of variables which may not be available in real time deployment. For instance, some variables were observed to have high correlations, such as ladle silicon content and our target of silicon content in hot metal. It was ascertained that these variables were auxiliary variables produced during the laboratory sample analysis rather than input parameters or furnace state variables. Thus, such variables were also removed.

Finally, correlation tests are used in a similar but simpler manner to Cui’s [4] work which uses Gray Relational Analysis (GRA) to obtain corresponding time lags for each of the features. These correlations are with respect to silicon content output. For instance, Figure 8 shows that coke rate seems to have a time lag of around four casts (2.5 h per cast), while the more immediate factors such as natural gas injection have no time lag, i.e., they affect the current cast. Thus, the focus is kept on time periods along the lag times corresponding to the furnace parameters.

#### 2.2.6. Feature Engineering and Standardization

After thoroughly preprocessing the data, it is processed into a form suitable for generic ML models input by steps such as standardization and feature reduction and selection. For standardization, Robust Scalers for numerical inputs and Ordinal Encoders (relevant to XGBoost inputs as opposed to One-Hot Encoders) for categorical inputs are used, implemented in python library scikit-learn (v 1.4.2). Feature selection is then performed via Principal Component Analysis (PCA) for certain numerical features and Multi-Correlation Analysis (MCA) for categorical features to reduce the number of features used. For simpler models, some feature engineering steps such as having polynomial features could be used but for a sufficiently complex ensemble model such as XGBoost, this step need not be used. A subset of these steps is described in Figure 9. Figure 9a shows a numerical value transform, while Figure 9b shows a categorical transform into an ordinal scale. Figure 9c utilizes MCA to concatenate seemingly unimportant categorical and meta-features together to form a relevant numerical vectorized encoding.

### 2.3. Modeling via XGBoost and Hyperparameter-Tuning-Based Selection

XGBoost [21] is utilized and experimented extensively for predictive modeling purposes. As an outline, XGBoost is an ensemble-based model with a Decision Tree as its basic unit. It works by assigning input data, such as all the operating and state conditions provided as the input features, with a particular category of outputs, or in our regression case, within an interval or cluster of expected Si content value. The final output for each tree is typically the mean of this interval or cluster. The model output is the aggregate of these decision trees. The splitting is achieved by observing the data features, composing the input data, and choosing which feature, at each part of the decision tree, provides the best split, as illustrated in Figure 10.

There are different heuristics in XGBoost, such as choosing from a subset of features and focusing on training for those data samples with greater probability for which the model is having a hard time predicting. This is performed for each decision tree, referred to as a weak learner, separately, and the randomization ensures that each tree learns to predict using different criteria, thereby reducing overfitting to some degree. For the final predictions, the mode (in the case of categories) or the average (in the case of regression) of the values produced by the learners in this ensemble is used. The XGBoost model is very suitable for tabular, structured data, especially when the amount of data is scarce. It provides a nice interpretation of input variables, unlike deep learning counterparts, which are more of a black box in this regard. XGBoost can also provide faster inference speed on such structured data due to the deterministic nature of the path traversal for their trees during prediction.

With the type of ML model determined, hyperparameter optimization is performed using a combination of random grid search and Bayesian Hyperparameter optimization to efficiently identify coarse ranges of probable hyperparameter values. This is followed by a more focused grid search within these refined intervals. By splitting the search process into these sequential steps, the trade-off between model accuracy and computational efficiency of the selection process is balanced. As a result, the model with the highest predictive accuracy among the candidate models is obtained.

## 3. Results

### 3.1. Processed Dataset for Evaluation

The data processing pipeline formats, appropriately filters, performs relevant feature selections on, and standardizes the data. This yields a dataset with about 1800 samples. Specifically, these samples contain the compressed information of the furnace condition and state pertaining to the respective cast. Each casting period takes approximately 2.5 h, and thus, the dataset spans roughly six months of standard furnace operations. The comprehensiveness in both the quality of furnace information contained in the dataset and the time span of the operations makes it suitable for performing tests and evaluations. For this purpose, a train–test split of 85–15%, or 1560 vs. 240 samples, via random selection is performed to obtain our corresponding training and test sets.

### 3.2. Accuracy

On the test set, using prior shift-based data allows this approach to achieve 91% accuracy, or 0.065% expected error with respect to actual silicon content%. Concomitantly, a fivefold cross validation is performed, due to the relatively low number of samples, which yields an accuracy in a tight range of 87% to 92% for these five folds. This validates the findings by reducing the likelihood of an overfitted model or the possibility of using a ‘lucky’ test split which fortuitously yielded good results.

Moreover, this accuracy is also a lower bound of the model’s accuracy, as only the past shift’s aggregated data—which have a guarantee of availability—is utilized to predict the next shift’s silicon content output. Further improvements in accuracy requiring minimal model changes might be achieved by incorporating real-time features. While other features such as hot metal chemistry have a lag, these real-time features from sensor data such as blast moisture, blast temperature, coke rate, and natural gas injection rates can readily improve performance. This is reflected in the feature importance score for such features in Table 2. Simultaneously, some of these features exhibit high variability in real time, the information of which may not be known to the model at that time, and thus can impact prediction accuracy. Hence, these same limitations hold the key to improving performance but may be rendered difficult to achieve due to furnace data processing and provisioning logistics in real-time. Apart from the model accuracy, with potential for improvement, it is also computationally feasible on low-spec machines. The model occupies minimal memory space, 1.1 MB, and can provide a prediction in less than 1 ms.

The accuracy of 91% or expected error bound of 0.065% in silicon content with 0.911% mean is highly promising. It is also reasonable to ascertain that a good portion of these gains can be attributed to the data processing pipeline. This assertion is corroborated in Figure 11, where the accuracy of this approach is compared with those obtained from other models, such as sequential neural network models or SVM models. Cases are also studied without applying the data processing pipeline—only with basic formatting and data clean-up. It can be observed that even the more complex neural network model’s accuracy is lower than our Bayesian optimized XGBoost model, where hyperparameter tuning depends on the quality of processed data. Additionally, the pipeline results are compared with a commercial neural network modeling software, Neuroshell (v 3.0.0.1), which has inbuilt data processing and hyperparameter optimization. In this case, the proposed model’s accuracy remains superior. This suggests that in an industrial setting with hundreds of parameters, where manual selection becomes infeasible or is limiting, the selection pipeline can help identify the best reduced features and yield a better predictive model.

Alongside internal comparisons, this model’s results and accuracy are in line with observed prediction accuracy and error bounds seen in other literature. For instance, Song’s [22], Cui’s [4], and Diniz’s [5] work exhibit an error bound of 0.13% (mean silicon content 0.5%), 0.1252% (mean silicon content 0.55%), and 0.05 (mean silicon content 0.45%), respectively, against our 0.065% (mean silicon content 0.911%). Direct comparisons are not possible because individual datasets in the studies are not publicly available, in part due to the possibility of disclosure of sensitive industrial data. Nevertheless, the approach detailed herein holds promise. The generality and adaptability of the pipeline makes it suitable for diverse industrial process data; for data refinement; and for attaining accurate models for the purpose for modeling various operational states, outputs, or variables of interest besides hot metal silicon [18,23].

### 3.3. Robustness and Reliability

Furthermore, additional result analysis was conducted to determine robustness and interpretability. The histogram plot in Figure 12 shows the error deviation and that the residuals are approximately centered. This indicates a lack of bias in the predictions which is typically a good indicator in terms of overfitting. However, the lack of a large test set means that outliers are highlighted slightly more. The cross-validation accuracy, which is consistently around the 90–91% mark, serves to reduce this effect and provides further evidence of robustness.

Complementing this, the time series plot in Figure 13, provides another perspective on the model’s performance. Specifically, predictions are performed on approximately 220 samples from the test set. The inputs to the model for each sample are the processed furnace parameters and state variables available from the previous cast. The models in turn predict the silicon content of the current cast. The model’s predictions align closely with the actual reported silicon content %, especially including the peaks and troughs of silicon content production. The values reported are standardized and can hence be negative. The standardized silicon content has a linear relation with the actual silicon content and thus does not affect the evaluation metric scores such as the accuracy percentage.

The reliance on historical training data means the model may require periodic retraining to address data staleness. This need can also arise due to changes in furnace operation parameters or structural modifications. To maintain accuracy and robustness, continuous evaluation of the model is essential, focusing on amortized accuracy over time rather than isolated periods of poor performance. When a consistent decline in accuracy or input data drift is observed, definitive action can be taken. Operators can provide real-time feedback on poorly performing cases, enabling the model to adapt by adding subtrees to handle these examples. However, given the low resource intensity of the current model, retraining on the entire dataset in a batch setting is generally more advisable. Techniques such as sample weighting can also be employed to prioritize critical cases during retraining.

### 3.4. Interpretability

Interpretability in XGBoost, unlike typical deep learning models, is one of its desired qualities, especially in industrial settings where there is a crucial need to understand the actual factors impacting production quantity rather than simply predicting the quantity itself. To achieve this, XGBoost also provides an internal ranking of feature importance or “gain scores” for each input parameter. Gain in XGBoost measures the improvement in loss reduction—how far off predictions are from ground truth silicon content during training—due to a feature split, illustrated in Figure 10, without directly accounting for interactions with other features. It is also pertinent to note that these scores are not linear in nature. Mathematically, these are the “gains” during XGBoost training where each feature based split yields improved entropy and helps determine if that feature was useful for fitting the model on the output silicon values in the train set. Likewise, the model does not have a concept of domain knowledge, so these features should be seen as complex correlation factors rather than actual causation factors.

In Table 3, a mix of such variables can be observed. For instance, cast average titanium is not expected to have any causal relation with determining silicon content but it may be correlated in some way with the silicon content. The XGBoost model identifies this potential correlation. However, there are other features indicated by the model that align well with those established by theory and empirical observations as causal variables, such as hot blast temperature, moisture, and cast temperature. These are expected variables from physical principles that establish the link between predicted Si content in the hot metal and the available energy in the high energy zone of the furnace as noted in the Section 1.

Figure 14 shows the SHapley Additive exPlanation factor (SHAP), which is a highly utilized method for ascertaining impact of each feature and its interpretability. It measures the marginal contributions of a feature by considering all possible subsets of that feature. It thus provides another view of the “impact” of each feature, mainly the global standalone contribution of each feature. For instance, it can be observed that cast average titanium has a positive correlation with silicon content, but coke rate seems to only have an impactful positive correlation when its value is high. Low coke rate does not seem to lead to low silicon content, at least not with a similar impact. Natural gas injection shows an opposite effect compared to coke rate. Finally, we also see that “mca__0” feature, a compressed sum of categorical variables such as SiO_2_ content, does not have a high SHAP value despite the high gain value, which values critical splitting, in Table 3. This indicates that mca__0 might have critical importance in a small subset of splits for predictions, specifically the outliers with higher individual losses, and that it is highly correlated with other features hence diluting its standalone SHAP importance value.

### 3.5. Error Analysis

It is worthwhile to conduct an error analysis for the high error cases observed in this study, where a large difference is observed between the model’s predictions and actual silicon content, in order to assess potential avenues for further improvement. It can also reveal properties regarding the nature of the operations, systematic biases, and stochasticity of the processes. Thus, the current test set is augmented with new samples and segregated based on the high error cases against lower error ones. The higher error cases are defined as those that exhibit a difference of twice the mean expected error, which is 0.13%. These come out to be approximately 9.1% of the cases, or 107 cases from the augmented 1180 sized dataset.

Afterwards, the feature characteristics of the five of top important features and the silicon content are visualized. Figure 15 visualizes these characteristics in the form of box plots and the corresponding kernel density estimates (KDEs). The box plot serves to show data statistics such as median, interquartile range, and outliers, while the KDE shows a non-parametric probability density function (pdf), essentially the estimated distribution, of the relevant features. It is observed that the natural gas injection rate has a higher mean value with a lower standard deviation, while cast average temperature has a higher standard deviation with a lower mean for higher error cases. However, the shape of their pdfs overall is consistent, and no definitive pattern emerges to conclude otherwise. The lack of difference in the distribution of the features and silicon output is somewhat unexpected but still plausible given the lack of systematic bias seen before.

Therefore, a time series-based analysis was conducted to observe if higher errors occur in local clusters. There were indeed definitive patterns in this case, as illustrated in Figure 16 where sample actual silicon and model predictions for higher error and lower error cases in successive casts are juxtaposed. Higher error predictions generally arise when there is a lot of local variability, noise and oscillations in silicon content production across successive casts. This could indicate high unexplained stochasticity and noise in the operations of the furnace during the relevant period, not captured by the furnace parameters. Thus, the model struggles to provide a good prediction for those samples. Stochasticity in processes is inherent however their explainability can be improved by potentially including other information impacting the furnace state not yet covered by the current parameters. This includes data such as inter-shaft pressure differentials and estimates for liquid level and fraction in the hearth from other analysis approaches. Overall, the findings seem to indicate lack of systemic biases in the model and that high errors are likely caused by stochasticity in the furnace operations during localized periods.

## 4. Conclusions

In this paper, a comprehensive, generalized data preprocessing pipeline integrated with a robust machine learning (ML) modeling framework to predict blast furnace (BF) hot metal silicon content was developed and demonstrated. Using the XGBoost model, 91% accuracy and 0.065% expected error in silicon content were achieved for next-cast average silicon content predictions at run speeds faster than 1 ms, even with relatively stale and historical input data. This ensures data availability for potential deployments without reliance on real-time inputs. Furthermore, robustness analysis, including knowing when and how to retrain the model, and error analysis confirmed the reliability of the model. These aspects enable operators to accurately gauge silicon content and make informed adjustments to manage energy consumption and mitigate operational risks. Additionally, by controlling silicon effectively, operators can also reduce energy consumption and potentially carbon emissions in downstream processes in steelmaking, including the Basic Oxygen Furnace (BOF).

The pipeline’s generality across industrial facilities is intrinsic to the relative consistency of the raw data schema provided by industrial collaborators for integrated steelmaking operations. Typical furnace data processing systems, such as Pi or similar platforms, usually associate furnace parameters or state variables either with a timestamp (e.g., gas rate at a specific time for a particular furnace) or, in more processed cases, with operational events (e.g., cast average zinc content during a casting or ladle process). By specifying linking keys such as “timestamp” or “cast” in the pipeline’s user configuration, the full state of the furnace—including input variables and the output variable to model—can be effectively determined. Our approach is agnostic to the specific data source if the required linking criteria are met. The pipeline can then handle diverse data sources while enabling user-configurable settings for anomaly thresholds (e.g., thermocouple temperature), linking variables (e.g., casts or timestamps), and output variables (e.g., silicon content). Additionally, it dynamically performs adaptive correlations and feature selection based on the modeled output variables [18,23]. Moreover, the pipeline can be easily extended to predict other industrial process variables, such as slag chemistry, with minimal modifications. This adaptability enables rapid development of bespoke models tailored to different sites and operational requirements [18,23], making our approach both efficient and versatile.

Despite the lack of standardized data for benchmarking, the pipeline developed in this study represents a significant advancement. The ability to seamlessly integrate features on an industrial scale while remaining flexible to changing process variables is as yet undocumented in the current literature for ironmaking applications. This novelty not only enhances prediction accuracy but also offers adaptability in creating and selecting bespoke models, providing a pathway for future advancements in predictive modeling for industrial processes.

An evident area for enhancing the predictions generated by this pipeline is the integration of a powerful deep learning sequential model to incorporate time-based information. By extending the approach to yield extended forecasts and not merely short-term predictions, blast furnace operators can be offered valuable insights for longer-term planning and stability. Additionally, incorporating real-time process data into the current model—which currently relies on historical data—could further elevate the model’s capability. This would enable more accurate real-time silicon content predictions, empowering operators to make swift adjustments for immediate operational changes. This enhanced schema, considering the inherent complexity and stochasticity of blast furnace operations, likely holds the potential to deliver the most accurate and actionable predictions.

## Figures and Tables

**Figure 1 materials-18-00632-f001:**
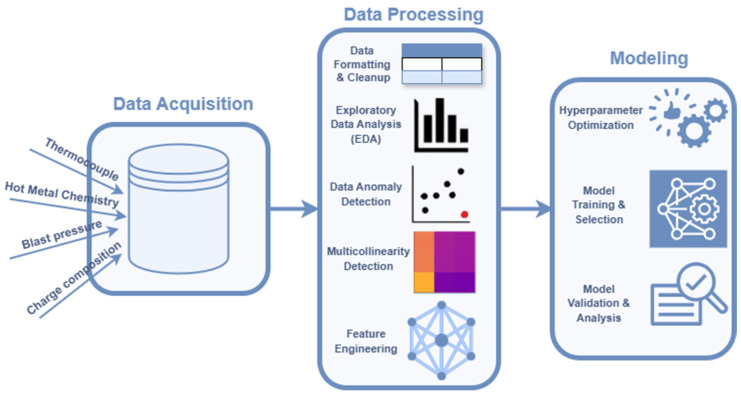
Key processing and modeling steps delineating a generalized pipeline adaptable to furnace-specific data processing.

**Figure 2 materials-18-00632-f002:**
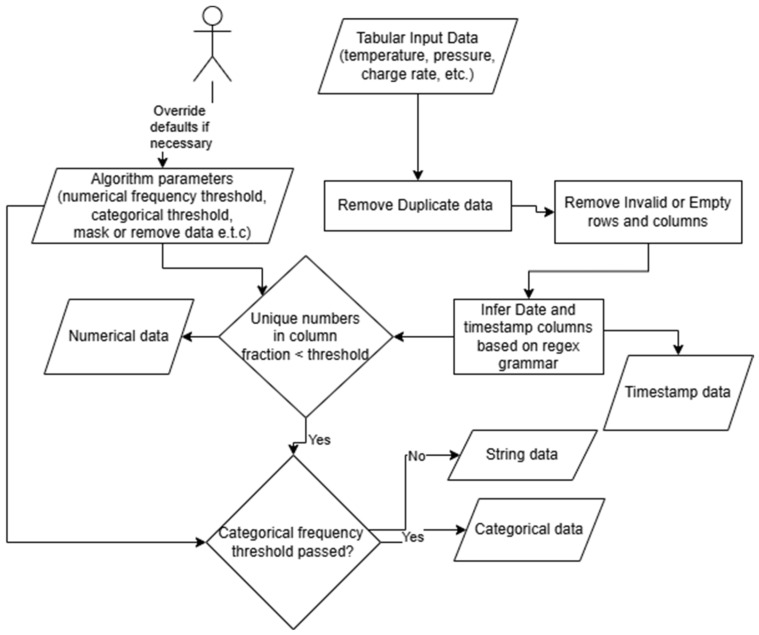
Data type inference via regex, unique count, and configurable thresholds.

**Figure 3 materials-18-00632-f003:**
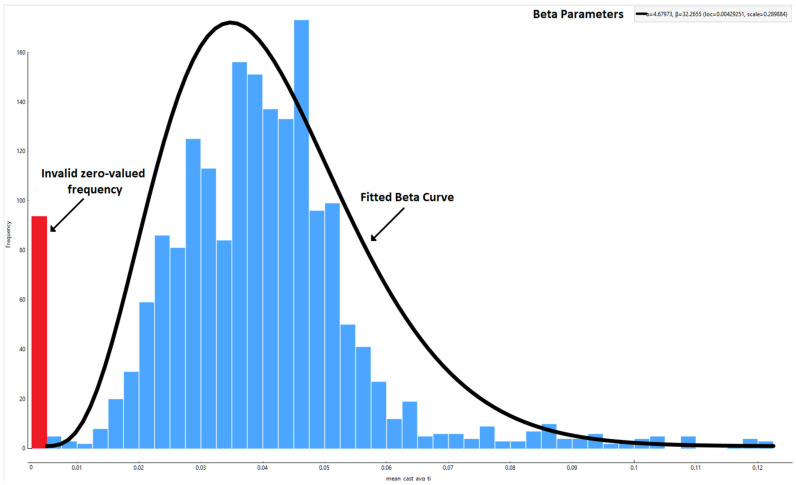
Histogram of a feature skewed around 0, coupled with beta curve fitting as a guide, indicating invalid or unknown titanium content values rather than the absence of titanium.

**Figure 4 materials-18-00632-f004:**
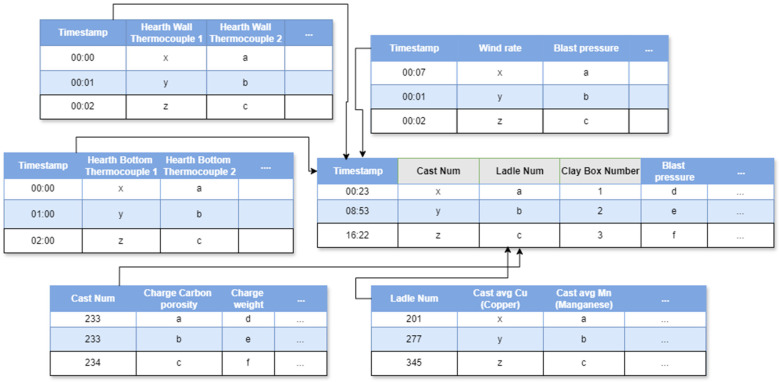
Linkage among tables in our data source, with gray-marked non-influencing or linking columns removed downstream manually or via VIF.

**Figure 5 materials-18-00632-f005:**
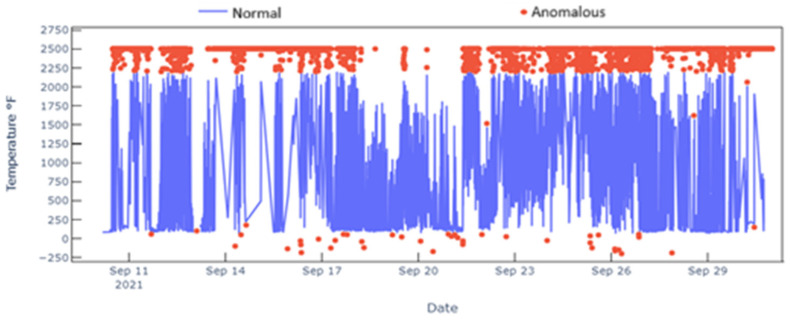
Time series plot of a thermocouple with anomalous readings detected in red.

**Figure 6 materials-18-00632-f006:**
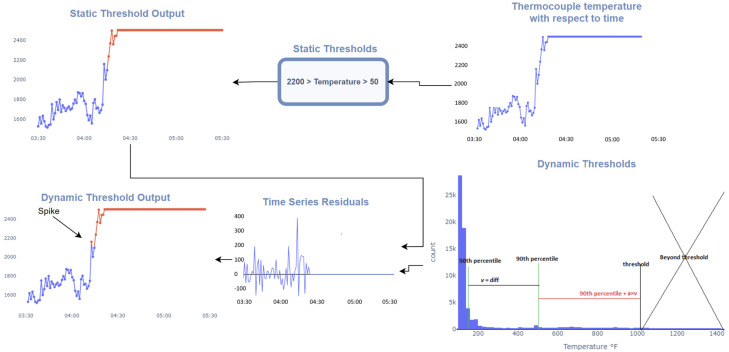
Static and dynamic thresholds identify out-of-range readings and spikes (red).

**Figure 7 materials-18-00632-f007:**
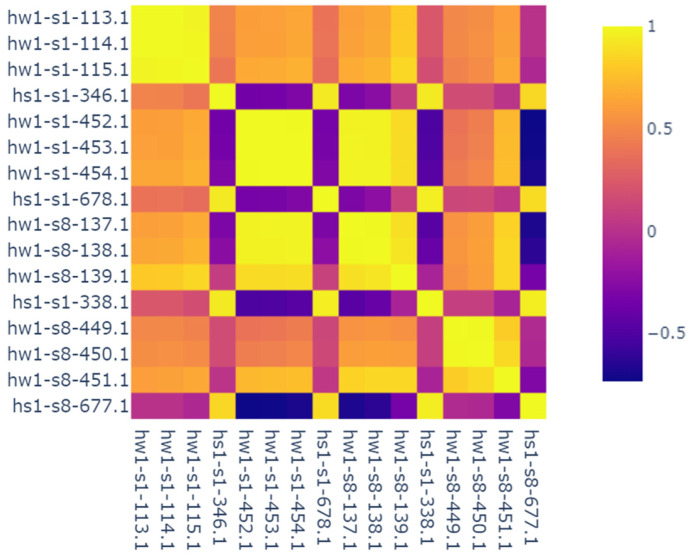
Correlation map indicating high correlations between proximal thermocouple readings.

**Figure 8 materials-18-00632-f008:**
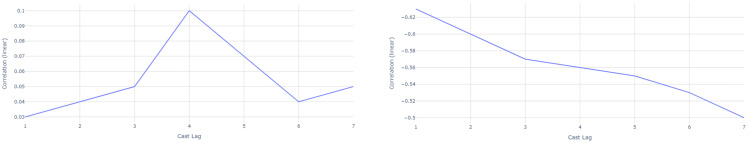
Coke rate (**left**) and natural gas injection rate (**right**) illustrating lag factors of 4 casts and 1 cast (current input cast) respectively.

**Figure 9 materials-18-00632-f009:**
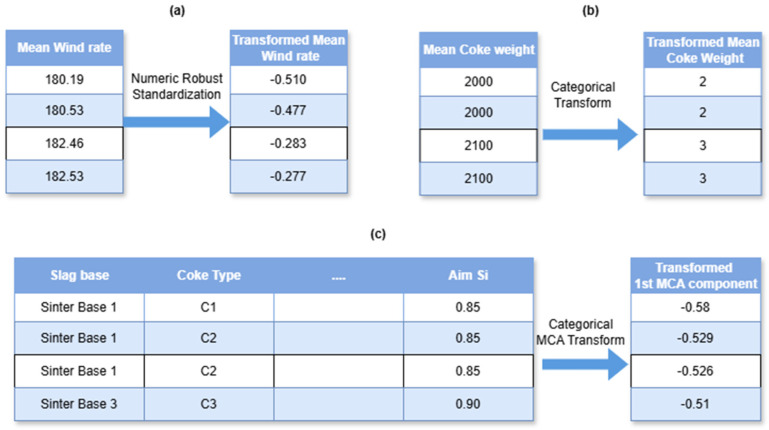
Representation of data standardization via (**a**) a numerical transform, (**b**) a categorical-to-ordinal transform, and (**c**) an MCA transform with one component shown.

**Figure 10 materials-18-00632-f010:**
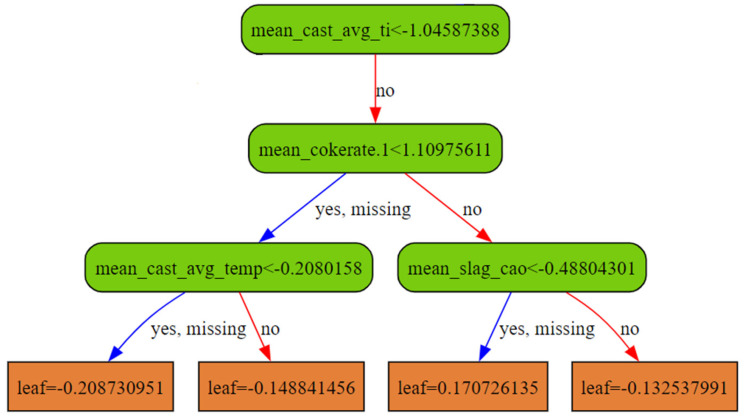
Subsection of a decision tree from the XGB ensemble, illustrating feature splits and standardized silicon content predictions.

**Figure 11 materials-18-00632-f011:**
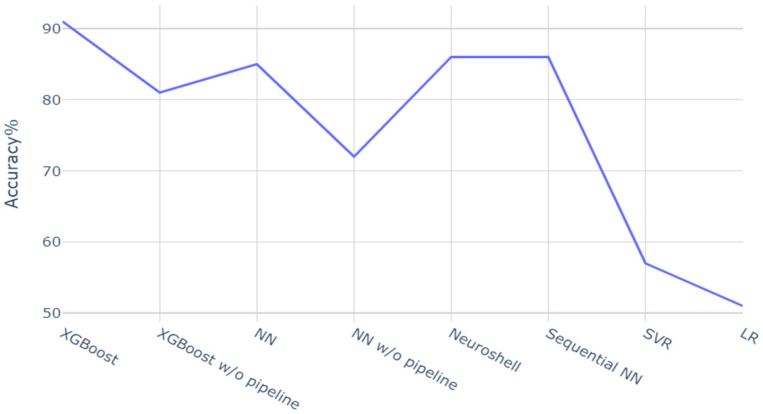
Relative accuracy of different models, such as XGBoost, Neural Network (NN), Support Vector Regression (SVR), and Linear Regression (LR), and the commercial Neuroshell software coupled with our data processing pipeline.

**Figure 12 materials-18-00632-f012:**
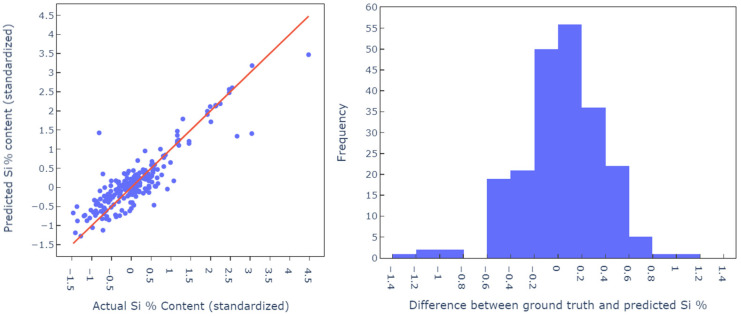
Parity plot and residual error distribution between predicted and actual silicon content, with equivariance, indicating minimal prediction bias.

**Figure 13 materials-18-00632-f013:**
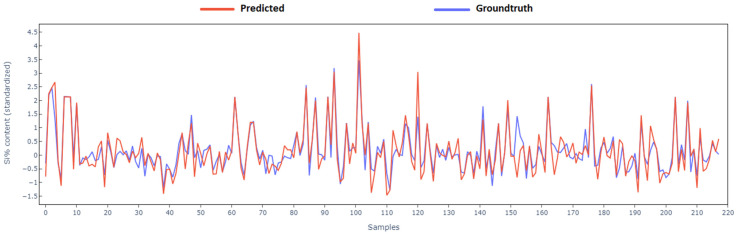
XGBoost model predictions compared to real (normalized) Si% content on randomly chosen but temporally sorted cast samples.

**Figure 14 materials-18-00632-f014:**
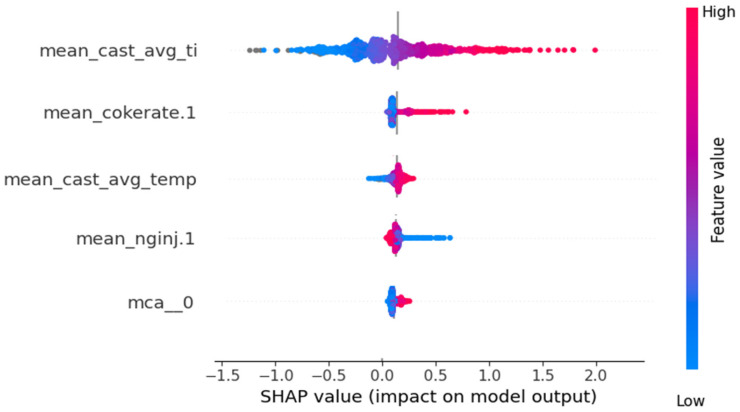
SHAP values for select important features indicating their impact on the model’s output and the individual contribution of the features.

**Figure 15 materials-18-00632-f015:**
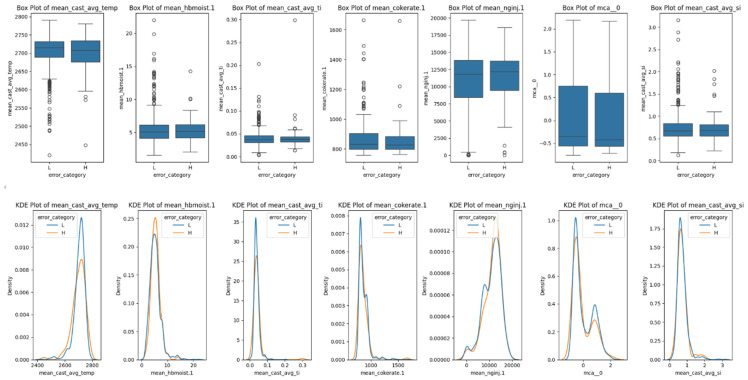
Box plots and KDEs of the most important features and silicon output, grouped by higher and lower modeling predictive error.

**Figure 16 materials-18-00632-f016:**
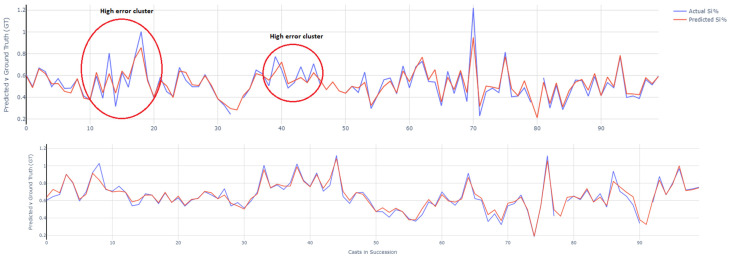
Higher (**top**) and lower (**bottom**) model predictive error cases. Higher errors are observed across noisy and oscillating local periods of silicon content production.

**Table 1 materials-18-00632-t001:** Basic data statistics of a few furnace parameters and states and output silicon content, representative of the types of features of the furnace.

Parameters	Unit	Range	Mean	Standard Deviation	Sampling Frequency	Missing
Thermocouple temperature (sample hearth walls)	°F	[378, 505]	457	16	Per minute	4%
Thermocouple temperature (sample hearth bottom)	°F	[671, 1455]	1017	154	Per hour (avg)	0%
Snort Valve Position	% (valve open)	[3.20, 98.3]	95.8	12.6	Per minute	0%
Top Gas CO_2_	%	[3.76, 20.8]	18.8	1.67	Per minute	0%
Slag SiO_2_	% (in slag)	[10.1, 57.4]	38.5	2.31	Per cast	11%
Wind rate	kscfm	[5.53, 201]	186	11.2	Per minute	0%
Cast Avg Temperature	°F	[2432, 2844]	2716	40.7	Per cast	3%
Hot Metal Temperature	°F	[1787, 2818]	2713	57.8	Per cast	0%
Coke rate	lbm/thm	[868, 1671]	937	88.6	Per minute	0%
Cast Avg Titanium	% (in hot metal)	[0.005, 0.122]	0.041	0.0156	Per cast	5%
Cast Avg Manganese	% (in hot metal)	[0.280, 0.417]	0.331	0.0334	Per cast	0%
Cast Avg Copper	% (in hot metal)	[0.001, 0.092]	0.005	0.00423	Per cast	5%
Top Gas Temperature	°F	[100, 677]	239	55.5	Per minute	0%
Top Gas Pressure	psi	[0.4, 26.4]	19.4	3.37	Per minute	0%
Hot Blast Moisture	grains/scf	[1.54, 22.1]	6.35	2.42	Per minute	0%
Hot Blast Pressure	psi	[0.05, 41.8]	35.8	5.11	Per minute	0%
Hot Blast Temperature	°F	[902, 2220]	2134	88	Per minute	0%
Slag CaO	% (in slag)	[34.2, 40.5]	39.0	0.872	Per cast	0%
Natural Gas Injection Rate	scfm	[2.42, 20,000]	11200	3800	Per minute	0%
Silicon content	% (in hot metal)	[0.288, 3.407]	0.911	0.283	Per cast	5%

**Table 2 materials-18-00632-t002:** Data type inference results from different sources.

Data Feature	Samples					Inferred Type
Hot metal Temperature	2704	2381	2099	2442	2387	Number
Bauxite	1800	2000	2000	1800	1800	Categorical (Number)
Thermocouple Timestamp	21/1/2021 00:14:00	27/4/2022 07:14:00	22/10/2021 22:19:00	17/7/2021 (day start)	7/8/2020	Date
Cast Number	2	3181	1772	222	399	Meta/String
Taphole Number	1	1	3	3	2	Categorical (Integer)

**Table 3 materials-18-00632-t003:** Feature importance in the XGBoost model, including known factors such as hot metal temperature and unexpected ones such as mean moisture and SiO_2_ content.

Feature Name	Description	Score (Relative)
mca_0	Sum of categorical columns, with the most important being SiO_2_ being provisioned to the furnace	669
mean_hbmoist.1	Mean Hot Blast Moisture	293
mean_cast_avg_temp	Mean cast average temperature	282
mean_hb_temp	Mean hot blast temperature	266
mean_cast_avg_ti	Mean content of titanium in cast	247
mean_hm_temp	Mean hot metal temperature in the ladle	233
mean_tg_co2.1	Selected top gas analyzer: Mean CO_2_ content in top gas	231
mean_nginj.1	The rate of natural gas injection	225
mean_cokerate.1	The rate of coke (carbon) charged	208

## Data Availability

The original contributions presented in this study are included in the article. Further inquiries can be directed to the corresponding author.

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
