# Peer review of "Prediction of Silicon Content in a Blast Furnace via Machine Learning: A Comprehensive Processing and Modeling Pipeline"

_materials, 2025, doi:10.3390/ma18030632_

Round 1

Reviewer 1 Report

Comments and Suggestions for Authors

The manuscript proposes a generalized data processing scheme to adapt to different furnace parameters and modeling to establish a machine learning (ML) model capable of predicting the Si content of hot metal with reasonable accuracy. The presented method predicts the average Si content of the upcoming furnace casting with 91% accuracy based on 200 target predictions for a specific furnace provided by the Xgboost model.

In order to improve the manuscript, I propose the following changes:

1) The introduction should be supplemented by a short description of the scope of the manuscript presenting the individual activities of the authors

2) In subsection 1, the authors state that they received the test data from the steel manufacturer, listed them but did not provide any parametric data of their scope and multiplicity of use for experiments. It is therefore necessary to supplement it by a tabular presentation of the range of parameters used in the methodology

3) Fig. 1. It is good but a mathematical notation is also necessary to present the key steps of data processing and modeling

4) Table 1 must be supplemented with sample data and their values. The authors must provide examples of course, selected, but the reader must be given an introductory scope

5) In subsection 2.2, the authors must present the mathematical application of the Gamma distribution they refer to

6) For formatting and cleaning, a graphical algorithm for these operations should be presented

7) In subsection 2.4, the role of filters and models should be explained in detail, and a graphical sample of their operation should be presented

Reviewer 2 Report

Comments and Suggestions for Authors

Dear authors,

I have several comments on your work (see attached file).

Please clarify and add any additional information.

Best regards

Reviewer 3 Report

Comments and Suggestions for Authors

The dataset used for testing is not described in detail (e.g., size, representativeness). Clarify the size of the test dataset, and if possible, perform cross-validation to ensure robustness.

The limitations of the current model (e.g., reliance on historical data, lack of real-time feedback) are briefly mentioned but not analyzed in depth. 

The scalability of the model to different furnace configurations or datasets has not been thoroughly explored. Include comments on how adaptable the pipeline is to datasets from other facilities with different operational profiles.

The discussion lacks a detailed comparison of the findings with results from similar studies cited in the manuscript. 

While operational benefits are discussed, economic impacts are not quantified.

Streamline the introduction by focusing on the specific research problem, objectives, and contributions while reducing historical context.

Include statistical validation techniques (e.g., k-fold cross-validation) to ensure robustness and generalizability of results.

Add a comparative analysis against simpler models (e.g., Support Vector Regression, Linear Regression) or deep learning approaches.

Conduct a detailed error analysis, including case studies or examples of predictions where the model performed poorly.

Round 2

Reviewer 1 Report

Comments and Suggestions for Authors

The authors of the manuscript have improved my suggestions that I presented in the review and have significantly improved the substantive level of their research. I accept the manuscript in its presented form and consider it ready for publication. A small remark concerns the description of tables that should be placed above the table.

Reviewer 2 Report

Comments and Suggestions for Authors

Vážený autor,

ďakujem za prijatie a zapracovanie mojich predchádzajúcich komentárov. Nemám ďalšie pripomienky.

S pozdravom